# Learning to Reconstruct Shapes from Unseen Classes

**Xiuming Zhang**\*
MIT CSAIL

**Zhoutong Zhang**\*
MIT CSAIL

**Chengkai Zhang**
MIT CSAIL

**Joshua B. Tenenbaum**
MIT CSAIL

**William T. Freeman**
MIT CSAIL, Google Research

**Jiajun Wu**
MIT CSAIL

## Abstract

From a single image, humans are able to perceive the full 3D shape of an object by exploiting learned shape priors from everyday life. Contemporary single-image 3D reconstruction algorithms aim to solve this task in a similar fashion, but often end up with priors that are highly biased by training classes. Here we present an algorithm, *Generalizable Reconstruction (GenRe)*, designed to capture more generic, class-agnostic shape priors. We achieve this with an inference network and training procedure that combine 2.5D representations of visible surfaces (depth and silhouette), spherical shape representations of both visible and non-visible surfaces, and 3D voxel-based representations, in a principled manner that exploits the causal structure of how 3D shapes give rise to 2D images. Experiments demonstrate that GenRe performs well on single-view shape reconstruction, and generalizes to diverse novel objects from categories not seen during training.

## 1 Introduction

Humans can imagine an object's full 3D shape from just a single image, showing only a fraction of the object's surface. This applies to common objects such as chairs, but also to novel objects that we have never seen before. Vision researchers have long argued that the key to this ability may be a sophisticated hierarchy of representations, extending from images through surfaces to volumetric shape, which process different aspects of shape in different representational formats [Marr, 1982]. Here we explore how these ideas can be integrated into state-of-the-art computer vision systems for 3D shape reconstruction.

Recently, computer vision and machine learning researchers have made impressive progress on single-image 3D reconstruction by learning a parametric function $f_{2D\rightarrow3D}$, implemented as deep neural networks, that maps a 2D image to its corresponding 3D shape. Essentially, $f_{2D\rightarrow3D}$ encodes shape priors ("what realistic shapes look like"), often learned from large shape repositories such as ShapeNet [Chang et al., 2015]. Because the problem is well-known to be ill-posed—there exist many 3D explanations for any 2D visual observation—modern systems have explored looping in various structures into this learning process. For example, MarrNet [Wu et al., 2017] uses intrinsic images or 2.5D sketches [Marr, 1982] as an intermediate representation, and concatenates two learned mappings for shape reconstruction: $f_{2D\rightarrow3D} = f_{2.5D\rightarrow3D} \circ f_{2D\rightarrow2.5D}$.

Many existing methods, however, ignore the fact that mapping a 2D image or a 2.5D sketch to a 3D shape involves complex, but deterministic geometric projections. Simply using a neural network to approximate this projection, instead of modeling this mapping explicitly, leads to inference models that are overparametrized (and hence subject to overfitting training classes). It also misses valuable inductive biases that can be wired in through such projections. Both of these factors contribute to poor generalization to unseen classes.

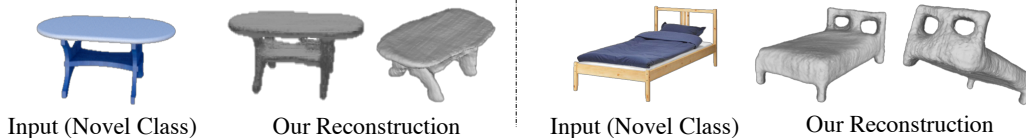

| Input (Novel Class) | Our Reconstruction | Input (Novel Class) | Our Reconstruction |

Figure 1: We study the task of generalizable single-image 3D reconstruction, aiming to reconstruct the 3D shape of an object outside training classes. Here we show a table and a bed reconstructed from single RGB images by our model trained on cars, chairs, and airplanes. Our model learns to reconstruct objects outside the training classes.

Here we propose to disentangle geometric projections from shape reconstruction to better generalize to unseen shape categories. Building upon the MarrNet framework [Wu et al., 2017], we further decompose $f_{2.5D \to 3D}$ into a deterministic geometric projection $p$ from 2.5D to a partial 3D model, and a learnable completion $c$ of the 3D model. A straightforward version of this idea would be to perform shape completion in the 3D voxel grid: $f_{2.5D \to 3D} = c_{3D \to 3D} \circ p_{2.5D \to 3D}$. However, shape completion in 3D is challenging, as the manifold of plausible shapes is sparser in 3D than in 2D, and empirically this fails to reconstruct shapes well.

Instead we perform completion based on spherical maps. Spherical maps are surface representations defined on the UV coordinates of a unit sphere, where the value at each coordinate is calculated as the minimal distance travelled from this point to the 3D object surface along the sphere's radius. Such a representation combines appealing features of 2D and 3D: spherical maps are a form of 2D images, on which neural inpainting models work well; but they have a semantics that allows them to be projected into 3D to recover full shape geometry. They essentially allow us to complete non-visible object surfaces from visible ones, as a further intermediate step to full 3D reconstruction. We now have $f_{2.5D \to 3D} = p_{S \to 3D} \circ c_{S \to S} \circ p_{2.5D \to S}$, where $S$ stands for spherical maps.

Our full model, named *Generalizable Reconstruction (GenRe)*, thus comprises three cascaded, learnable modules connected by fixed geometric projections. First, a single-view depth estimator predicts depth from a 2D image ($f_{2D \to 2.5D}$); the depth map is then projected into a spherical map ($p_{2.5D \to S}$). Second, a spherical map inpainting network inpaints the partial spherical map ($c_{S \to S}$); the inpainted spherical map is then projected into 3D voxels ($p_{2.5D \to 3D}$). Finally, we introduce an additional voxel refinement network to refine the estimated 3D shape in voxel space. Our neural modules only have to model object geometry for reconstruction, without having to learn geometric projections. This enhances generalizability, along with several other factors: during training, our modularized design forces each module of the network to use features from the previous module, instead of directly memorizing shapes from the training classes; also, each module only predicts outputs that are in the same domain as its inputs (image-based or voxel-based), which leads to more regular mappings.

Our GenRe model achieves state-of-the-art performance on reconstructing shapes both within and outside training classes. Figure 1 shows examples of our model reconstructing a table and a bed from single images, after training only on cars, chairs, and airplanes. We also present detailed analyses of how each component contributes to the final prediction.

This paper makes three contributions. First, we emphasize the task of generalizable single-image 3D shape reconstruction. Second, we propose to disentangle geometric projections from shape reconstruction, and include spherical maps with differentiable, deterministic projections in an integrated neural model. Third, we demonstrate that the resulting model achieves state-of-the-art performance on single-image 3D shape reconstruction for objects within and outside training classes.

## 2 Related Work

**Single-image 3D reconstruction.** The problem of recovering the object shape from a single image is challenging, as it requires both powerful recognition systems and prior knowledge of plausible 3D shapes. Large CAD model repositories [Chang et al., 2015] and deep networks have contributed to the significant progress in recent years, mostly with voxel representations [Choy et al., 2016, Girdhar et al., 2016, Häne et al., 2017, Kar et al., 2015, Novotny et al., 2017, Rezende et al., 2016, Tatarchenko et al., 2016, Tulsiani et al., 2017, Wu et al., 2016, 2017, 2018, Zhu et al., 2018, Yan et al., 2016]. Apart from voxels, some researchers have also studied reconstructing objects in point clouds [Fan et al., 2017] or octave trees [Riegler et al., 2017, Tatarchenko et al., 2017]. The shape

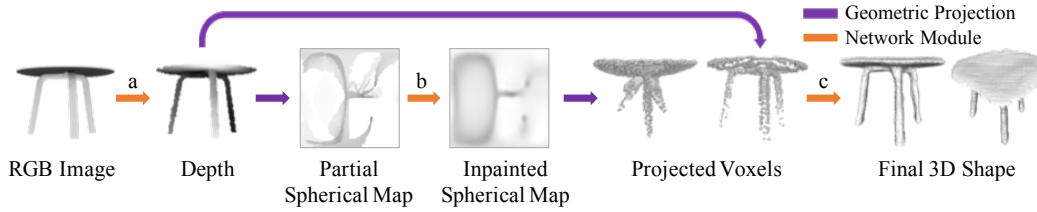

Figure 2: Our model for generalizable single-image 3D reconstruction (GenRe) has three components: (a) a depth estimator that predicts depth in the original view from a single RGB image, (b) a spherical inpainting network that inpaints a partial, single-view spherical map, and (c) a voxel refinement network that integrates two backprojected 3D shapes (from the inpainted spherical map and from depth) to produce the final output.

priors learned in these approaches, however, are in general only applicable to their training classes, with very limited generalization power for reconstructing shapes from unseen categories. In contrast, our system exploits 2.5D sketches and spherical representations for better generalization to objects outside training classes.

**Spherical projections.**    Spherical projections have been shown effective in 3D shape retrieval [Esteves et al., 2018], classification [Cao et al., 2017], and finding possible rotational as well as reflective symmetries [Kazhdan et al., 2004, 2002]. Recent papers [Cohen et al., 2018, 2017] have studied differentiable, spherical convolution on spherical projections, aiming to preserve rotational equivariance within a neural network. These designs, however, perform convolution in the spectral domain with limited frequency bands, causing aliasing and loss of high-frequency information. In particular, convolution in the spectral domain is not suitable for shape reconstruction, since the reconstruction quality highly depends on the high-frequency components. In addition, the ringing effects caused by aliasing would introduce undesired artifacts.

**2.5D sketch recovery.**    The origin of intrinsic image estimation dates back to the early years of computer vision [Barrow and Tenenbaum, 1978]. Through years, researchers have explored recovering 2.5D sketches from texture, shading, or color images [Barron and Malik, 2015, Bell et al., 2014, Horn and Brooks, 1989, Tappen et al., 2003, Weiss, 2001, Zhang et al., 1999]. As handy depth sensors get mature [Izadi et al., 2011], and larger-scale RGB-D datasets become available [McCormac et al., 2017, Silberman et al., 2012, Song et al., 2017], many papers start to estimate depth [Chen et al., 2016, Eigen and Fergus, 2015], surface normals [Bansal and Russell, 2016, Wang et al., 2015], and other intrinsic images [Janner et al., 2017, Shi et al., 2017] with deep networks. Our method employs 2.5D estimation as a component, but focuses on reconstructing shapes from unseen categories.

**Zero- and few-shot recognition.**    In computer vision, abundant attempts have been made to tackle the problem of few-shot recognition. We refer readers to the review article [Xian et al., 2017] for a comprehensive list. A number of earlier papers have explored sharing features across categories to recognize new objects from a few examples [Bart and Ullman, 2005, Farhadi et al., 2009, Lampert et al., 2009, Torralba et al., 2007]. More recently, many researchers have begun to study zero- or few-shot recognition with deep networks [Akata et al., 2016, Antol et al., 2014, Hariharan and Girshick, 2017, Wang et al., 2017, Wang and Hebert, 2016]. Especially, Peng et al. [2015] explored the idea of learning to recognize novel 3D models via domain adaptation.

While these proposed methods are for recognizing and categorizing images or shapes, in this paper we explore reconstructing the 3D shape of an object from unseen classes. This problem has received little attention in the past, possibly due to its considerable difficulty. A few imaging systems have attempted to recover 3D shape from a single shot by making use of special cameras [Proesmans et al., 1996, Sagawa et al., 2011]. Unlike them, we study 3D reconstruction from a single RGB image. Very recently, researchers have begun to look at the generalization power of 3D reconstruction algorithms [Shin et al., 2018, Jayaraman et al., 2018, Rock et al., 2015, Funk and Liu, 2017]. Here we present a novel approach that makes use of spherical representations for better generalization.

# 3  Approach

Single-image reconstruction algorithms learn a parametric function $f_{\text{2D}\rightarrow\text{3D}}$ that maps a 2D image to a 3D shape. We tackle the problem of generalization by regularizing $f_{\text{2D}\rightarrow\text{3D}}$. The key regularization we impose is to factorize $f_{\text{2D}\rightarrow\text{3D}}$ into geometric projections and learnable reconstruction modules.

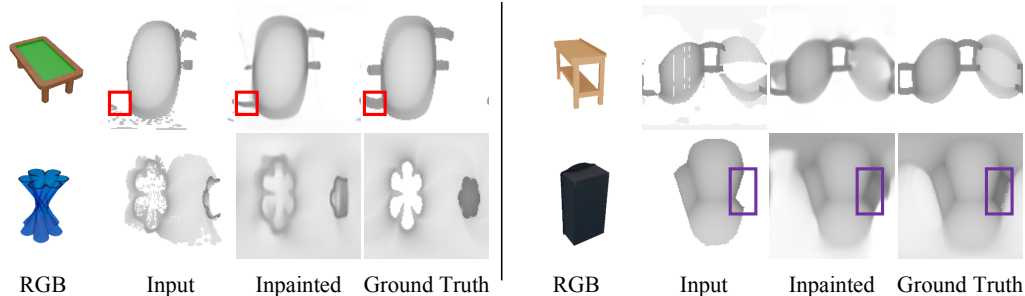

| RGB | Input | Inpainted | Ground Truth | RGB | Input | Inpainted | Ground Truth |

Figure 3: Examples of our spherical inpainting module generalizing to new classes. Trained on chairs, cars, and planes, the module completes the partially visible leg of the table (red boxes) and the unseen cabinet bottom (purple boxes) from partial spherical maps projected from ground-truth depth.

Our GenRe model consists of three learnable modules, connected by geometric projections as shown in Figure 2. The first module is a single-view depth estimator $f_{2D\rightarrow2.5D}$ (Figure 2a), taking a color image as input and estimates its depth map. As the depth map can be interpreted as the visible surface of the object, the reconstruction problem becomes predicting the object's complete surface given this partial estimate.

As 3D surfaces are hard to parametrize efficiently, we use spherical maps as a surrogate representation. A geometric projection module ($p_{2.5D\rightarrow S}$) converts the estimated depth map into a spherical map, referred to as the partial spherical map. It is then passed to the spherical map inpainting network ($c_{S\rightarrow S}$, Figure 2b) to predict an inpainted spherical map, representing the object's complete surface. Another projection module ($p_{S\rightarrow 3D}$) projects the inpainted spherical map back to the voxel space.

As spherical maps only capture the outermost surface towards the sphere, they cannot handle self-occlusion along the sphere's radius. We use a voxel refinement module (Figure 2c) to tackle this problem. It takes two 3D shapes as input, one projected from the inpainted spherical map and the other from the estimated depth map, and outputs a final 3D shape.

## 3.1 Single-View Depth Estimator

The first component of our network predicts a depth map from an image with a clean background. Using depth as an intermediate representation facilitates the reconstruction process by distilling essential geometric information from the input image [Wu et al., 2017].

Further, depth estimation is a class-agnostic task: shapes from different classes often share common geometric structure, despite distinct visual appearances. Take beds and cabinets as examples. Although they are of different anatomy in general, both have perpendicular planes and hence similar patches in their depth images. We demonstrate this both qualitatively and quantitatively in Section 4.4.

## 3.2 Spherical Map Inpainting Network

With spherical maps, we cast the problem of 3D surface completion into 2D spherical map inpainting. Empirically we observe that networks trained to inpaint spherical maps generalize well to new shape classes (Figure 3). Also, compared with voxels, spherical maps are more efficient to process, as 3D surfaces are sparse in nature; quantitatively, as we demonstrate in Section 4.5 and Section 4.6, using spherical maps results in better performance.

As spherical maps are signals on the unit sphere, it is tempting to use network architectures based on spherical convolution [Cohen et al., 2018]. They are however not suitable for our task of shape reconstruction. This is because spherical convolution is conducted in the spectral domain. Every conversion to and from the spectral domain requires capping the maximum frequency, causing extra aliasing and information loss. For tasks such as recognition, the information loss may be negligible compared with the advantage of rotational invariance offered by spherical convolution. But for reconstruction, the loss leads to blurred output with only low-frequency components. We empirically find that standard convolution works much better than spherical convolution under our setup.

## 3.3 Voxel Refinement Network

Although an inpainted spherical map provides a projection of an object's surface onto the unit sphere, the surface information is lost when self-occlusion occurs. We use a refinement network that operates

in the voxel space to recover the lost information. This module takes two voxelized shapes as input, one projected from the estimated depth map and the other from the inpainted spherical map, and predicts the final shape. As the occluded regions can be recovered from local neighboring regions, this network only needs to capture local shape priors and is therefore class-agnostic. As shown in the experiments, when provided with ground-truth depth and spherical maps, this module performs consistently well across training and unseen classes.

## 3.4 Technical Details

**Single-view depth estimator.** Following Wu et al. [2017], we use an encoder-decoder network for depth estimation. Our encoder is a ResNet-18 [He et al., 2015], encoding a $256 \times 256$ RGB image into 512 feature maps of size $1 \times 1$. The decoder is a mirrored version of the encoder, replacing all convolution layers with transposed convolution layers. In addition, we adopt the U-Net structure [Ronneberger et al., 2015] and feed the intermediate outputs of each block of the encoder to the corresponding block of the decoder. The decoder outputs the depth map in the original view at the resolution of $256 \times 256$. We use an $\ell_2$ loss between predicted and target images.

**Spherical map inpainting network.** The spherical map inpainting network has a similar architecture as the single-view depth estimator. To reduce the gap between standard and spherical convolutions, we use periodic padding to both inputs and training targets in the longitude dimension, making the network aware of the periodic nature of spherical maps.

**Voxel refinement network.** Our voxel refinement network takes as input voxels projected from the estimated, original-view depth and from the inpainted spherical map, and recovers the final shape in voxel space. Specifically, the encoder takes as input a two-channel $128 \times 128 \times 128$ voxel (one for coarse shape estimation and the other for surface estimation), and outputs a 320-D latent vector. In decoding, each layer takes an extra input directly from the corresponding level of the encoder.

**Geometric projections.** We make use of three geometric projections: a depth to spherical map projection, a depth map to voxel projection, and a spherical map to voxel projection. For the depth to spherical map projection, we first convert depth into 3D point clouds using camera parameters, and then turn them into surfaces with the marching cubes algorithm [Lewiner et al., 2003]. Then, the spherical representation is generated by casting rays from each UV coordinate on the unit sphere to the sphere's center. This process is not differentiable. To project depth or spherical maps into voxels, we first convert them into 3D point clouds. Then, a grid of voxels is initialized, where the value of each voxel is determined by the average distance between all the points inside it to its center. Then, for all the voxels that contain points, we negate its value and add it by 1. This projection process is fully differentiable.

**Training.** We train our network with viewer-centered 3D supervision, where the 3D shape is rotated to match the object's pose in the input image. This is in contrast to object-centered approaches, where the 3D supervision is always in a predefined pose regardless of the object's pose in the input image. Object-centered approaches are less suitable for reconstructing shapes from new categories, as predefined poses are unlikely to generalize across categories.

We first train the 2.5D sketch estimator with RGB images and their corresponding depth images, all rendered with ShapeNet [Chang et al., 2015] objects (see Section 4.2 and the supplemental material for details). We then train the spherical map inpainting network with single-view (partial) spherical maps and the ground-truth full spherical maps as supervision. Finally, we train the voxel refinement network on coarse shapes predicted by the inpainting network as well as 3D surfaces backprojected from the estimated 2.5D sketches, with the corresponding ground-truth shapes as supervision. We then jointly fine-tune the spherical inpainting module and the voxel refinement module with both 3D shape and 2D spherical map supervision.

## 4 Experiments

### 4.1 Baselines

We organize baselines based on the shape representation they use.

**Voxels.** Voxels are arguably the most common representation for 3D shapes in the deep learning era due to their amenability to 3D convolution. For this representation, we consider DRC [Tulsiani et al., 2017] and MarrNet [Wu et al., 2017] as baselines. Our model uses $128^3$ voxels of $[0, 1]$ occupancy.

**Mesh and point clouds.** Considering the cubic complexity of the voxel representation, recent papers have explored meshes [Groueix et al., 2018, Yao et al., 2018] and point clouds [Fan et al.,

2017] in the context of neural networks. In this work, we consider AtlasNet [Groueix et al., 2018] as a baseline.

**Multi-view maps.** Another way of representing 3D shapes is to use a set of multi-view depth images [Soltani et al., 2017, Shin et al., 2018, Jayaraman et al., 2018]. We compare with the model from Shin et al. [2018] in this regime.

**Spherical maps.** As introduced in Section 1, one can also represent 3D shapes as spherical maps. We include two baselines with spherical maps: first, a one-step baseline that predicts final spherical maps directly from RGB images (GenRe-1step); second, a two-step baseline that first predicts single-view spherical maps from RGB images and then inpaints them (GenRe-2step). Both baselines use the aforementioned U-ResNet image-to-image network architecture.

To provide justification for using spherical maps, we provide a baseline (3D Completion) that directly performs 3D shape completion in voxel space. This baseline first predicts depth from an input image; it then projects the depth map into the voxel space. A completion module takes the projected voxels as input and predicts the final result.

To provide a performance upper bound for our spherical inpainting and voxel refinement networks (b and c in Figure 2), we also include the results when our model has access to ground-truth depth in the original view (GenRe-Oracle) and to ground-truth full spherical maps (GenRe-SphOracle).

## 4.2 Data

We use ShapeNet [Chang et al., 2015] renderings for network training and testing. Specifically, we render each object in 20 random views. In addition to RGB images, we also render their corresponding ground-truth depth maps. We use Mitsuba [Jakob, 2010], a physically-based rendering engine, for all our renderings. Please see the supplementary material for details on data generation and augmentation.

For all models, we train them on the three largest ShapeNet classes (cars, chairs, and airplanes), and test them on the next 10 largest classes: bench, vessel, rifle, sofa, table, phone, cabinet, speaker, lamp, and display. Besides ShapeNet renderings, we also test these models, trained only on synthetic data, on real images from Pix3D [Sun et al., 2018], a dataset of real images and the ground-truth shape of every pictured object. In Section 5, we also test our model on non-rigid shapes such as humans and horses [Bronstein et al., 2008] and on highly regular shape primitives.

## 4.3 Metrics

Because neither depth maps nor spherical maps provide information inside shapes, our model predicts only surface voxels that are not guaranteed watertight. Consequently, intersection over union (IoU) cannot be used as an evaluation metric. We hence evaluate reconstruction quality using Chamfer distance (CD) [Barrow et al., 1977], defined as

$$\text{CD}(S_1, S_2) = \frac{1}{|S_1|} \sum_{x \in S_1} \min_{y \in S_2} \|x - y\|_2 + \frac{1}{|S_2|} \sum_{y \in S_2} \min_{x \in S_1} \|x - y\|_2, \tag{1}$$

where $S_1$ and $S_2$ are sets of points sampled from surfaces of the 3D shape pair. For models that output voxels, including DRC and our GenRe model, we sweep voxel thresholds from 0.3 to 0.7 with a step size of 0.05 for isosurfaces, compute CD with 1,024 points sampled from all isosurfaces, and report the best average CD for each object class.

Shin et al. [2018] reports that object-centered supervision produces better reconstructions for objects from the training classes, whereas viewer-centered supervision is advantaged in generalizing to novel classes. Therefore, for DRC and AtlasNet, we train each network with both types of supervision. Note that AtlasNet, when trained with viewer-centered supervision, tends to produce unstable predictions that render CD meaningless. Hence, we only present CD for the object-centered AtlasNet.

## 4.4 Results on Depth Estimation

We show qualitative and quantitative results on depth estimation quality across categories. As shown in Figure 4, our depth estimator learns effectively the concept of near and far, generalizes well to unseen categories, and does not show statistically significant deterioration as the novel test class gets increasingly dissimilar to the training classes, laying the foundation for the generalization power of our approach. Formally, the dissimilarity from test class $C_{\text{test}}$ to training classes $C_{\text{train}}$ is defined as $(1/|C_{\text{test}}|) \sum_{x \in C_{\text{test}}} \min_{y \in C_{\text{train}}} \text{CD}(x, y)$.

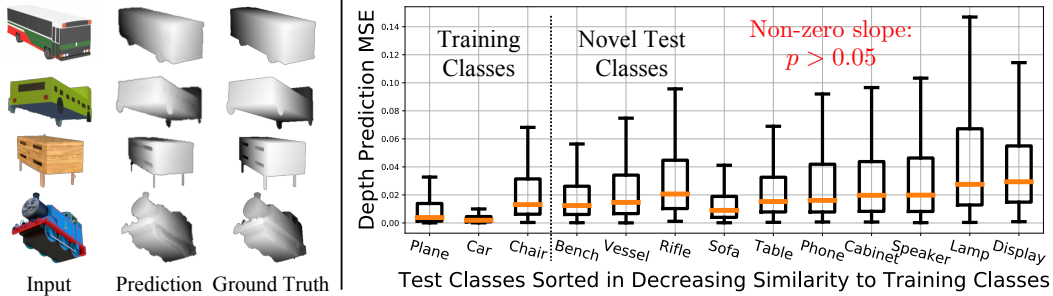

Figure 4: Left: Our single-view depth estimator, trained on cars, chairs, and airplanes, generalizes to novel classes: buses, trains, and tables. Right: As the novel test class gets increasingly dissimilar to the training classes (left to right), depth prediction does not show statistically significant degradation ($p > 0.05$).

| | Models | Seen | Unseen | | | | | | | | | | |
|---|---|---|---|---|---|---|---|---|---|---|---|---|---|
| | | | Bch | Vsl | Rfl | Sfa | Tbl | Phn | Cbn | Spk | Lmp | Dsp | Avg |
| Object-Centered | DRC [Tulsiani et al., 2017] | .072 | .112 | .100 | .104 | .108 | .133 | .199 | .168 | .164 | .145 | .188 | .142 |
| | AtlasNet [Groueix et al., 2018] | **.059** | .102 | **.092** | **.088** | .098 | .130 | .146 | .149 | .158 | .131 | .173 | .127 |
| Viewer-Centered | DRC [Tulsiani et al., 2017] | .092 | .120 | .109 | .121 | .107 | .129 | .132 | .142 | .141 | .131 | .156 | .129 |
| | MarrNet [Wu et al., 2017] | .070 | .107 | .094 | .125 | .090 | .122 | .117 | .125 | .123 | .144 | .149 | .120 |
| | Multi-View [Shin et al., 2018] | .065 | .092 | **.092** | .102 | .085 | .105 | .110 | .119 | .117 | .142 | .142 | .111 |
| | 3D Completion | .076 | .102 | .099 | .121 | .095 | .109 | .122 | .131 | .126 | .138 | .141 | .118 |
| | GenRe-1step | .063 | .104 | .093 | .114 | .084 | .108 | .121 | .128 | .124 | .126 | .151 | .115 |
| | GenRe-2step | .061 | .098 | .094 | .117 | .084 | .102 | .115 | .125 | .125 | **.118** | **.118** | .110 |
| | GenRe (Ours) | .064 | **.089** | **.092** | .112 | **.082** | **.096** | **.107** | **.116** | **.115** | .124 | .130 | **.106** |
| | GenRe-Oracle | .045 | .050 | .048 | .031 | .059 | .057 | .054 | .076 | .077 | .060 | .060 | .057 |
| | GenRe-SphOracle | .034 | .032 | .030 | .021 | .044 | .038 | .037 | .044 | .045 | .031 | .040 | .036 |

Table 1: Reconstruction errors (in CD) of the training classes and 10 novel classes, ordered from the most to the least similar to the training classes. Our model is viewer-centered by design, but achieves performance on par with the object-centered state of the art (AtlasNet) in reconstructing the seen classes. As for generalization to novel classes, our model outperforms the state of the art across 9 out of the 10 classes.

## 4.5 Reconstructing Novel Objects from Training Classes

We present results on generalizing to novel objects from the training classes. All models are trained on cars, chairs, and airplanes, and tested on unseen objects from the same three categories.

As shown in Table 1, our GenRe model is the best-performing viewer-centered model. It also outperforms most object-centered models except AtlasNet. GenRe's preformance is impressive given that object-centered models tend to perform much better on objects from seen classes [Shin et al., 2018]. This is because object-centered models, by exploiting the concept of canonical views, actually solve an easier problem. The performance drop from object-centered DRC to viewer-centered DRC supports this empirically. However, for objects from unseen classes, the concept of canonical views is no longer well-defined. As we will see in Section 4.6, this hurts the generalization power of object-centered methods.

## 4.6 Reconstructing Objects from Unseen Classes

We study how our approach enables generalization to novel shape classes unseen during training.

**Synthetic renderings.** We use the 10 largest ShapeNet classes other than chairs, cars, and airplanes as our test set. Table 1 shows that our model consistently outperforms the state of the art, except for the class of rifles, in which AtlasNet performs the best. Qualitatively, our model produces reconstructions that are much more consistent with input images, as shown in Figure 5. In particular, on unseen classes, our results still attain good consistency with the input images, while the competitors either lack structural details present in the input (e.g., 5) or retrieve shapes from the training classes (e.g., 4, 6, 7, 8, 9).

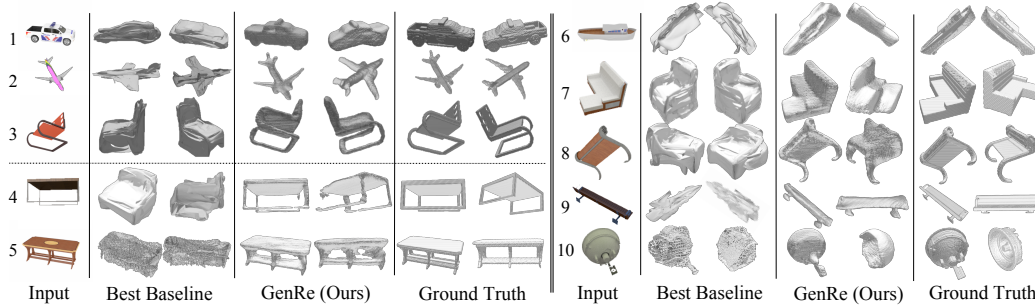

| | | | | | | | | | |
|1| | | | |6| | | | |
|2| | | | |7| | | | |
|3| | | | |8| | | | |
|4| | | | |9| | | | |
|5| | | | |10| | | | |

Input    Best Baseline    GenRe (Ours)    Ground Truth    Input    Best Baseline    GenRe (Ours)    Ground Truth

Figure 5: Single-image 3D reconstructions of objects within and beyond training classes. Each row from left to right: the input image, two views from the best-performing baseline for each testing object (1-4, 6-9: AtlasNet; 5, 10: Shin et al. [2018]), two views of our GenRe predictions, and the ground truth. All models are trained on the same dataset of cars, chairs, and airplanes.

Comparing our model with its variants, we find that the two-step approaches (GenRe-2step and GenRe) outperform the one-step approach across all novel categories. This empirically supports the advantage of our two-step modeling strategy that disentangles geometric projections from shape reconstruction.

**Real images.**    We further compare how our model, AtlasNet, and Shin et al. [2018] perform on real images from Pix3D. Here, all models are trained on ShapeNet cars, chairs, and airplanes, and tested on real images of beds, bookcases, desks, sofas, tables, and wardrobes.

Quantitatively, Table 2 shows that our model outperforms the two competitors across all novel classes except beds, for which Shin et al. [2018] performs the best. For chairs, one of the training classes, the object-centered AtlasNet leverages the canonical view and outperforms the two viewer-centered approaches. Qualitatively, our reconstructions preserve the details present in the input (e.g., the hollow structures in the second row of Figure 6).

| | AtlasNet | Shin et al. | GenRe |
|---|---|---|---|
| Chair | **.080** | .089 | .093 |
| Bed | .114 | **.106** | .113 |
| Bookcase | .140 | .109 | **.101** |
| Desk | .126 | .121 | **.109** |
| Sofa | .095 | .088 | **.083** |
| Table | .134 | .124 | **.116** |
| Wardrobe | .121 | .116 | **.109** |

Table 2: Reconstruction errors (in CD) for seen (chairs) and unseen classes (the rest) on real images from Pix3D. GenRe outperforms the two baselines across all unseen classes except beds. For chairs, object-centered AtlasNet performs the best by leveraging the canonical view.

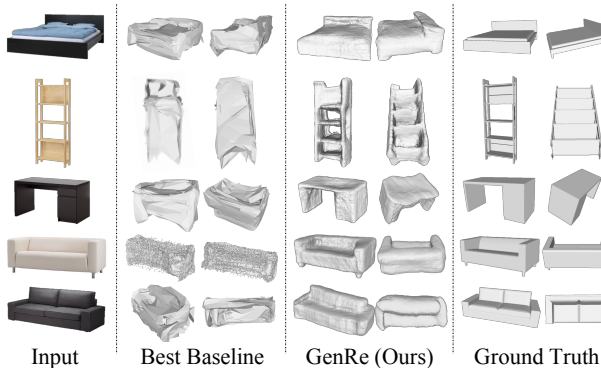

Input    Best Baseline    GenRe (Ours)    Ground Truth

Figure 6: Reconstructions on real images from Pix3D by GenRe and AtlasNet or Shin et al. [2018]. All models are trained on cars, chairs, and airplanes.

## 5   Analyses

### 5.1   The Effect of Viewpoints on Generalization

The generic viewpoint assumption states that the observer is not in a special position relative to the object [Freeman, 1994]. This makes us wonder if the "accidentalness" of the viewpoint affects the quality of reconstructions.

As a quantitative analysis, we test our model trained on ShapeNet chairs, cars, and airplanes on 100 randomly sampled ShapeNet tables, each rendered in 200 different views sampled uniformly on a sphere. We then compute, for each of the 200 views, the median CD of the 100 reconstructions. Finally, in Figure 7, we visualize these median CDs as a heatmap over an elevation-azimuth view

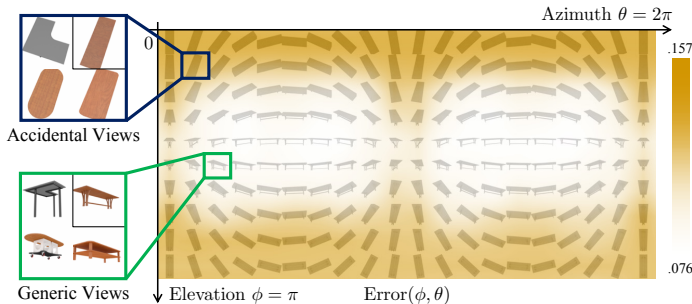

Figure 7: Reconstruction errors (CD) for different input viewpoints. The vertical (horizontal) axis represents elevation (azimuth). Accidental views (blue box) lead to large errors, while generic views (green box) result in smaller errors. Errors are computed for 100 tables; these particular tables are for visualization purposes only.

grid. As the heatmap shows, our model makes better predictions when the input view is generic than when it is accidental, consistent with our intuition.

## 5.2 Reconstructing Non-Rigid Shapes

We probe the generalization limit of our model by testing it with unseen non-rigid shapes, such as horses and humans. As the focus is mainly on the spherical map inpainting network (Figure 2b) and the voxel refinement network (Figure 2c), we assume our model has access to the ground-truth single-view depth (i.e., GenRe-Oracle) in this experiment. As demonstrated in Figure 8, our model not only retains the visible details in the original view, but also completes the unseen surfaces using the generic shape priors learned from rigid objects (cars, chairs, and airplanes).

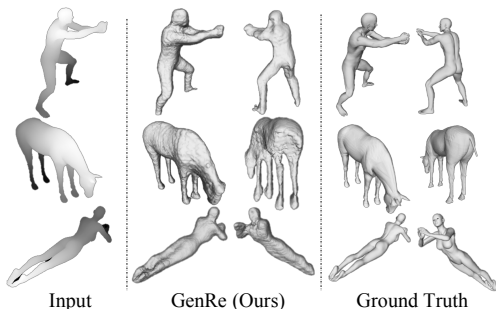

Figure 8: Single-view completion of non-rigid shapes from depth maps by our model trained on cars, chairs, and airplanes.

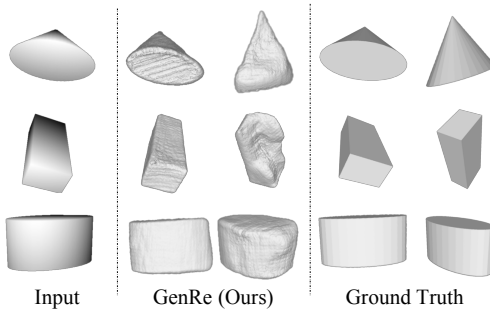

Figure 9: Single-view completion of highly regular shapes (primitives) from depth maps by our model trained on cars, chairs, and airplanes.

## 5.3 Reconstructing Highly Regular Shapes

We further explore whether our model captures global shape attributes by testing it on highly regular shapes that can be parametrized by only a few attributes (such as cones and cubes). Similar to Section 5.2, the model has only seen cars, chairs, and airplanes during training, and we assume our model has access to the ground-truth single-view depth (i.e., GenRe-Oracle).

As Figure 9 shows, although our model hallucinates the unseen parts of these shape primitives, it fails to exploit global shape symmetry to produce correct predictions. This is not surprising given that our network design does not explicitly model such regularity. A possible future direction is to incorporate priors that facilitate learning high-level concepts such as symmetry.

## 6 Conclusion

We have studied the problem of generalizable single-image 3D reconstruction. We exploit various image and shape representations, including 2.5D sketches, spherical maps, and voxels. We have proposed GenRe, a novel viewer-centered model that integrates these representations for generalizable, high-quality 3D shape reconstruction. Experiments demonstrate that GenRe achieves state-of-the-art performance on shape reconstruction for both seen and unseen classes. We hope our system will inspire future research along this challenging but rewarding research direction.

**Acknowledgements** We thank the anonymous reviewers for their constructive comments. This work is supported by NSF #1231216, NSF #1447476, ONR MURI N00014-16-1-2007, Toyota Research Institute, Shell, and Facebook.

## Footnotes

\* indicates equal contribution. Project page: http://genre.csail.mit.edu

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
