[Supplementary Material · genre_nips_supp.pdf]

# Supplemental Material:
# Learning to Reconstruct Shapes from Unseen Classes

## A.1  Data Preparation

We describe how we prepare our data for network training and testing.

**Scene setup.**  The camera is fully specified by its azimuth and elevation angles, as its distance from the object is fixed at 2.2, its up vector is always the world $+y$ axis, and it always looks at the world origin, where the object center lies. Focal length of the camera is fixed at 50mm on a 35mm film. Depth values are measured from the camera center (i.e., ray depth), rather than from the image plane.

**Rendering.**  We render 20 images of random views (or 200 fixed views in the viewpoint study) for each object of interest. To boost the rendering realism and diversity, we use three types of background: the SUN backgrounds [Xiao et al., 2010], high-dynamic-range environment lighting crawled on the web, and pure white backgrounds. Specifically, for each rendering, we randomly sample a background type and then a random instance of that type. We use Mitsuba [Jakob, 2010] for all of our rendering.

**Data augmentation.**  For network training, we augment our RGB images with three techniques: color jittering, adding lighting noise, and color normalization. In color jittering, we multiply the brightness, contrast, and saturation, one by one in a random order, by a random factor uniformly sampled from $[0.6, 1.4]$. We then add AlexNet-style lighting noise [Krizhevsky et al., 2012] and perform the standard color normalization with statistics derived from the ImageNet dataset [Deng et al., 2009].

## A.2  Model Details

We implement all of our networks in PyTorch 0.3.

### A.2.1  Single-View Depth Estimator

We adopt an encoder-decoder architecture, where the encoder is a ResNet-18 [He et al., 2015] that encodes a 256×256 RGB image into 512 feature maps of size 1×1. Specifically, it consists of, in a sequential order,

```
Conv2d(3, 64, kernel=7, stride=2, pad=3)
BatchNorm2d(64, eps=1e-05, momentum=0.1)
ReLU(inplace)
MaxPool2d(kernel=3, stride=2, pad=1, dilation=1)
BasicBlock(
  (conv1): Conv2d(64, 64, kernel=3, stride=1, pad=1)
  (bn1): BatchNorm2d(64, eps=1e-05, momentum=0.1)
  (relu): ReLU(inplace)
  (conv2): Conv2d(64, 64, kernel=3, stride=1, pad=1)
  (bn2): BatchNorm2d(64, eps=1e-05, momentum=0.1)
)
BasicBlock(
  (conv1): Conv2d(64, 64, kernel=3, stride=1, pad=1)
  (bn1): BatchNorm2d(64, eps=1e-05, momentum=0.1)
  (relu): ReLU(inplace)
  (conv2): Conv2d(64, 64, kernel=3, stride=1, pad=1)
  (bn2): BatchNorm2d(64, eps=1e-05, momentum=0.1)
)
BasicBlock(
  (conv1): Conv2d(64, 128, kernel=3, stride=2, pad=1)
  (bn1): BatchNorm2d(128, eps=1e-05, momentum=0.1)
  (relu): ReLU(inplace)
  (conv2): Conv2d(128, 128, kernel=3, stride=1, pad=1)
  (bn2): BatchNorm2d(128, eps=1e-05, momentum=0.1)
  (downsample):
    Conv2d(64, 128, kernel=1, stride=2)
```

```
      BatchNorm2d(128, eps=1e-05, momentum=0.1)
)
BasicBlock(
  (conv1): Conv2d(128, 128, kernel=3, stride=1, pad=1)
  (bn1): BatchNorm2d(128, eps=1e-05, momentum=0.1)
  (relu): ReLU(inplace)
  (conv2): Conv2d(128, 128, kernel=3, stride=1, pad=1)
  (bn2): BatchNorm2d(128, eps=1e-05, momentum=0.1)
)
BasicBlock(
  (conv1): Conv2d(128, 256, kernel=3, stride=2, pad=1)
  (bn1): BatchNorm2d(256, eps=1e-05, momentum=0.1)
  (relu): ReLU(inplace)
  (conv2): Conv2d(256, 256, kernel=3, stride=1, pad=1)
  (bn2): BatchNorm2d(256, eps=1e-05, momentum=0.1)
  (downsample):
    Conv2d(128, 256, kernel=1, stride=2)
    BatchNorm2d(256, eps=1e-05, momentum=0.1)
)
BasicBlock(
  (conv1): Conv2d(256, 256, kernel=3, stride=1, pad=1)
  (bn1): BatchNorm2d(256, eps=1e-05, momentum=0.1)
  (relu): ReLU(inplace)
  (conv2): Conv2d(256, 256, kernel=3, stride=1, pad=1)
  (bn2): BatchNorm2d(256, eps=1e-05, momentum=0.1)
)
BasicBlock(
  (conv1): Conv2d(256, 512, kernel=3, stride=2, pad=1)
  (bn1): BatchNorm2d(512, eps=1e-05, momentum=0.1)
  (relu): ReLU(inplace)
  (conv2): Conv2d(512, 512, kernel=3, stride=1, pad=1)
  (bn2): BatchNorm2d(512, eps=1e-05, momentum=0.1)
  (downsample):
    Conv2d(256, 512, kernel=1, stride=2)
    BatchNorm2d(512, eps=1e-05, momentum=0.1)
)
BasicBlock(
  (conv1): Conv2d(512, 512, kernel=3, stride=1, pad=1)
  (bn1): BatchNorm2d(512, eps=1e-05, momentum=0.1)
  (relu): ReLU(inplace)
  (conv2): Conv2d(512, 512, kernel=3, stride=1, pad=1)
  (bn2): BatchNorm2d(512, eps=1e-05, momentum=0.1)
).
```

The decoder is a mirrored version of the encoder, with all convolution layers replaced by transposed convolution layers. Additionally, we adopt the U-Net structure [Ronneberger et al., 2015] by feeding the intermediate outputs of each encoder block to the corresponding decoder block. The decoder outputs an image of *relative* depth values in the original view at the same resolution as input. Specifically, the decoder comprises

```
RevBasicBlock(
  (deconv1): ConvTranspose2d(512, 256, kernel=3, stride=1, pad=1)
  (bn1): BatchNorm2d(256, eps=1e-05, momentum=0.1)
  (relu): ReLU(inplace)
  (deconv2): ConvTranspose2d(256, 256, kernel=3, stride=2, pad=1, out_pad=1)
  (bn2): BatchNorm2d(256, eps=1e-05, momentum=0.1)
  (upsample):
    ConvTranspose2d(512, 256, kernel=1, stride=2, out_pad=1)
    BatchNorm2d(256, eps=1e-05, momentum=0.1)
)
```

```
RevBasicBlock(
  (deconv1): ConvTranspose2d(256, 256, kernel=3, stride=1, pad=1)
  (bn1): BatchNorm2d(256, eps=1e-05, momentum=0.1)
  (relu): ReLU(inplace)
  (deconv2): ConvTranspose2d(256, 256, kernel=3, stride=1, pad=1)
  (bn2): BatchNorm2d(256, eps=1e-05, momentum=0.1)
)
RevBasicBlock(
  (deconv1): ConvTranspose2d(512, 128, kernel=3, stride=1, pad=1)
  (bn1): BatchNorm2d(128, eps=1e-05, momentum=0.1)
  (relu): ReLU(inplace)
  (deconv2): ConvTranspose2d(128, 128, kernel=3, stride=2, pad=1, out_pad=1)
  (bn2): BatchNorm2d(128, eps=1e-05, momentum=0.1)
  (upsample):
    ConvTranspose2d(512, 128, kernel=1, stride=2, out_pad=1)
    BatchNorm2d(128, eps=1e-05, momentum=0.1)
)
RevBasicBlock(
  (deconv1): ConvTranspose2d(128, 128, kernel=3, stride=1, pad=1)
  (bn1): BatchNorm2d(128, eps=1e-05, momentum=0.1)
  (relu): ReLU(inplace)
  (deconv2): ConvTranspose2d(128, 128, kernel=3, stride=1, pad=1)
  (bn2): BatchNorm2d(128, eps=1e-05, momentum=0.1)
)
RevBasicBlock(
  (deconv1): ConvTranspose2d(256, 64, kernel=3, stride=1, pad=1)
  (bn1): BatchNorm2d(64, eps=1e-05, momentum=0.1)
  (relu): ReLU(inplace)
  (deconv2): ConvTranspose2d(64, 64, kernel=3, stride=2, pad=1, out_pad=1)
  (bn2): BatchNorm2d(64, eps=1e-05, momentum=0.1)
  (upsample):
    ConvTranspose2d(256, 64, kernel=1, stride=2, out_pad=1)
    BatchNorm2d(64, eps=1e-05, momentum=0.1)
)
RevBasicBlock(
  (deconv1): ConvTranspose2d(64, 64, kernel=3, stride=1, pad=1)
  (bn1): BatchNorm2d(64, eps=1e-05, momentum=0.1)
  (relu): ReLU(inplace)
  (deconv2): ConvTranspose2d(64, 64, kernel=3, stride=1, pad=1)
  (bn2): BatchNorm2d(64, eps=1e-05, momentum=0.1)
)
RevBasicBlock(
  (deconv1): ConvTranspose2d(128, 64, kernel=3, stride=1, pad=1)
  (bn1): BatchNorm2d(64, eps=1e-05, momentum=0.1)
  (relu): ReLU(inplace)
  (deconv2): ConvTranspose2d(64, 64, kernel=3, stride=1, pad=1)
  (bn2): BatchNorm2d(64, eps=1e-05, momentum=0.1)
  (upsample):
    ConvTranspose2d(128, 64, kernel=1, stride=1)
    BatchNorm2d(64, eps=1e-05, momentum=0.1)
)
RevBasicBlock(
  (deconv1): ConvTranspose2d(64, 64, kernel=3, stride=1, pad=1)
  (bn1): BatchNorm2d(64, eps=1e-05, momentum=0.1)
  (relu): ReLU(inplace)
  (deconv2): ConvTranspose2d(64, 64, kernel=3, stride=1, pad=1)
  (bn2): BatchNorm2d(64, eps=1e-05, momentum=0.1)
)
ConvTranspose2d(128, 64, kernel=3, stride=2, pad=1, out_pad=1)
```

```
BatchNorm2d(64, eps=1e-05, momentum=0.1)
ReLU(inplace)
ConvTranspose2d(64, 1, kernel=8, stride=2, pad=3, out_pad=0).
```

Relative depth values provided by the predicted depth images are insufficient for conversions to spherical maps or voxels, as there are still two degrees of freedom undetermined: the minimum and maximum (or scale). Therefore, we have an additional branch decoding, also from the 512 feature maps, the minimum and maximum of the depth values. Specifically, it contains

```
Conv2d(512, 512, kernel=2, stride=2)
Conv2d(512, 512, kernel=4, stride=1)
ViewAsLinear()
Linear(in_features=512, out_features=256, bias=True)
BatchNorm1d(256, eps=1e-05, momentum=0.1)
ReLU(inplace)
Linear(in_features=256, out_features=128, bias=True)
BatchNorm1d(128, eps=1e-05, momentum=0.1)
ReLU(inplace)
Linear(in_features=128, out_features=2, bias=True).
```

Using the pretrained ResNet-18 as our network initialization, we then train this network with supervision on both the depth image (relative) and the minimum as well as maximum values. Under this setup, our network predicts effectively the absolute depth values of the input view, which allows us to project these depth values to the spherical representation or voxel grid.

This network was trained with a batch size of 4. We used Adam [Kingma and Ba, 2015] with a learning rate of $1e-3$, $\beta_1 = 0.5$, and $\beta_2 = 0.9$ for optimization.

### A.2.2   Spherical Map Inpainting Network

Our inpainting network shares the same architecture as the single-view depth estimator. To mimic the boundary conditions of spherical maps, we use replication padding for the vertical dimension (elevation) and periodic padding for the horizontal dimension (azimuth). The padding size is 16 for all dimensions.

This network was trained with a batch size of 4. We used Adam with a learning rate of $1e-4$, $\beta_1 = 0.5$, and $\beta_2 = 0.9$ for optimization.

### A.2.3   Voxel Refinement Network

Our voxel refinement network adopts the U-Net structure [Ronneberger et al., 2015] and uses a sequence of 3D convolution and transposed convolution layers. The input tensor is of shape BatchSize$\times 2 \times 128 \times 128 \times 128$, where one channel contains voxels projected from the predicted original-view depth map, and the other contains voxels projected from the inpainted spherical map. After fusion, the output tensor is of shape BatchSize$\times 1 \times 128 \times 128 \times 128$. Specifically, the network is sturctured as:

```
Unet(
    Conv3d_block(
        Conv3d(2, 20, kernel=8, stride=2, pad=3)
        BatchNorm3d(20, eps=1e-05, momentum=0.1)
        LeakyReLU(negative_slope=0.01)
    )
    Conv3d_block(
        Conv3d(20, 40, kernel=4, stride=2, pad=1)
        BatchNorm3d(40, eps=1e-05, momentum=0.1)
        LeakyReLU(negative_slope=0.01)
    )
    Conv3d_block(
        Conv3d(40, 80, kernel=4, stride=2, pad=1)
        BatchNorm3d(80, eps=1e-05, momentum=0.1)
        LeakyReLU(negative_slope=0.01)
    )
    Conv3d_block(
```

```
            Conv3d(80, 160, kernel=4, stride=2, pad=1)
            BatchNorm3d(160, eps=1e-05, momentum=0.1)
            LeakyReLU(negative_slope=0.01)
        )
        Conv3d_block(
            Conv3d(160, 320, kernel=4, stride=2, pad=1)
            BatchNorm3d(320, eps=1e-05, momentum=0.1)
            LeakyReLU(negative_slope=0.01)
        )
        Conv3d_block(
            Conv3d(320, 640, kernel=4, stride=1)
            BatchNorm3d(640, eps=1e-05, momentum=0.1)
            LeakyReLU(negative_slope=0.01)
        )
        full_conv_block(
            Linear(in_features=640, out_features=640, bias=True)
            LeakyReLU(negative_slope=0.01)
        )
        Deconv3d_skip(
            ConvTranspose3d(1280, 320, kernel=4, stride=1)
            BatchNorm3d(320, eps=1e-05, momentum=0.1)
            LeakyReLU(negative_slope=0.01)
        )
        Deconv3d_skip(
            ConvTranspose3d(640, 160, kernel=4, stride=2, pad=1)
            BatchNorm3d(160, eps=1e-05, momentum=0.1)
            LeakyReLU(negative_slope=0.01)
        )
        Deconv3d_skip(
            ConvTranspose3d(320, 80, kernel=4, stride=2, pad=1)
            BatchNorm3d(80, eps=1e-05, momentum=0.1)
            LeakyReLU(negative_slope=0.01)

        )
        Deconv3d_skip(
            ConvTranspose3d(160, 40, kernel=4, stride=2, pad=1)
            BatchNorm3d(40, eps=1e-05, momentum=0.1)
            LeakyReLU(negative_slope=0.01)
        )
        Deconv3d_skip(
            ConvTranspose3d(80, 20, kernel=8, stride=2, pad=3)
            BatchNorm3d(20, eps=1e-05, momentum=0.1)
            LeakyReLU(negative_slope=0.01)
        )
        Deconv3d_skip(
            ConvTranspose3d(40, 1, kernel=4, stride=2, pad=1)
        )
).
```

This network was trained with a batch size of 4. We used Adam with a learning rate of $1e{-}5$, $\beta_1 = 0.5$, and $\beta_2 = 0.9$ for optimization.