[Reviews · NeurIPS 2018]

Reviewer 1



This paper proposes a single-view 3D reconstruction method, designed for generalization. The proposed method is to first predict a 2.5D representation (depth-map) of the input image, compute its spherical projection, do inpainting on the spherical projection, and then finally predicting 3D voxel from this representation. The paper has extensive evaluation against other state-of-the-art methods and obtains competative quantiative results and visually impressive results. Strengths: - The paper is written very well and motivates the importance of generalization well. Each of it's design choices are well explained. - Results look nice and less of memorization. In particular, the results on the L-shaped couch is actually retaining its particular shape, as opposed to other methods that output a generic sofa. Small details such as legs are well preserved. - Evaluation against other methods are extensive, with ablative study. - Solving for the 3D shape in camera-aligned coordinate system makes a lot of sense since then the results are also aligned to the input image. I agree that this is the right way to solve the problem. - Usig the spherical projection representation is interesting. The strength lies in the fact that each module preserves the original image detail as much as possible and this makes sense as to why the final result does capture the shape of the actual input image. Weaknesses: - Limited number of qualitative results. There are only 8 results in the big Figure 5 and few other results in the figure. Particularly because CD error does not correlate well with the visual perception of the results, it would've been nice to see more results in the supplementary materials. It would also nice to show results from the depth-oracle. - One point is that the paper only shows results on man-made objects (only those shapes that are available in shapenet). It seems to generalize to these rigid, man-made objects but if the goal is really for generalization, the paper could do experiments on other types of 3D shapes such as non-rigid and deformable objects such as cloths/developable surfaces, humans and animals, which has different distribution of 3D shapes. For example, the method could be tested on 3D shaeps from the Tosca dataset (Bronstein et al.) which includes humans, horses, cats etc. Since these 3D shapes do not have any textures one could try with oracle-depth, adn oracle-sphereical depth map. - The ablative study does not have oracle-spherical depth map which would add more insight into it's error modes. In all I like this paper, it presents a simple and nice method through spherical projection. Adding further experiments with different object types other than man-made objects will further strengthen this paper. Post rebuttal: I stand-by my original rating. I thank the authors for adding more experiments, these extra results look very good.

Reviewer 2



The paper presents a method of single-image 3D reconstruction that works well also for unseen classes. The method is based on a sequence of networks/operations that 1) estimate the depth of the input object, 2) convert the depth map to a partial spherical map, 3) inpaint the spherical map, 4) refine the voxelized representation. These intermediate representations reduce the dependency in class-specific priors, allowing the system to generalize beyond the trained classes. Overall, I like the paper and I find the use of intermediate representations useful for the task of both intra-class reconstruction and the generalization to unknown classes. In particular, the use of spherical maps for representing shapes and having a network to inpaint the hidden parts is interesting. Another strong point of the paper is its presentation: the paper is well written and easy to follow and the experimental setup is well performed. In the weaknesses, the novelty in terms of network architecture seems limited as the method is using a sequence of standard blocks/losses. Also, I would expect to see more examples in the supp mat and applications on real images. I would like also to see more details why the different "Ours" variants perform that way for the "Train" case. Finally, including IoU for the voxel-based approaches will give a better insight in the comparisons. Also how well are the intermediate depths maps?

Reviewer 3



This paper propose a pipeline to perform view-centric single view 3D shape reconstruction. To avoid overfitting to the training data, the authors decompose the process in 3 steps, each of which is solved using a standard off the shelf deep network: (i) single view depth prediction (ii) spherical map (inferred from the depth) inpainting (iii) voxel (inferred from the inpainted spherical mal) completion. The authors claim their approach allows much better generalization than existing approaches. I like the general approach of the paper and find the quantitative results reasonably convincing, I thus clearly support acceptance. I however have a few comments/questions. Most importantly, I didn't find the experiments completely convincing since the only view-centric baseline is quite weak. I think it should be possible to re-train AltlasNet to perform view-centric perdictions (and even if it fails this is an interesting point), and thus have a stronger baseline. Some points are unclear to me: - How is the center of the object determined (especially from the depth map only)? How does it affect the results? - is there an alignment (e.g. ICP) before computing the CD? If not, I don't think the comparison with object centric approaches is fair, and results with alignment should be added. Some smaller points: - the different parts could easily be adapted so that the pipeline can be trained end-to-end. I think the paper claims it might limit generalization, but it would be good to actually see the results. - l 121-125 make claims about generalization to real images which don't really convince me. Pix3D (Pix3D: Dataset and Methods for Single-Image 3D Shape Modeling) would allow to test on real images, that would be a nice addition. - l. 126: I don't agree that depth prediction is class-agnostic